# Observation of transverse spin Nernst magnetoresistance induced by thermal spin current in ferromagnet/non-magnet bilayers

Dong-Jun Kim [1], Chul-Yeon Jeon[1], Jong-Guk Choi[1], Jae Wook Lee[1], Srivathsava Surabhi[2], Jong-Ryul Jeong[2], Kyung-Jin Lee [3,4] & Byong-Guk Park [1]

Electric generation of spin current via spin Hall effect is of great interest as it allows an efficient manipulation of magnetization in spintronic devices. Theoretically, pure spin current can be also created by a temperature gradient, which is known as spin Nernst effect. Here, we report spin Nernst effect-induced transverse magnetoresistance in ferromagnet/non-magnetic heavy metal bilayers. We observe that the magnitude of transverse magnetoresistance in the bilayers is significantly modified by heavy metal and its thickness. This strong dependence of transverse magnetoresistance on heavy metal evidences the generation of thermally induced pure spin current in heavy metal. Our analysis shows that spin Nernst angles of W and Pt have the opposite sign to their spin Hall angles. Moreover, our estimate implies that the magnitude of spin Nernst angle would be comparable to that of spin Hall angle, suggesting an efficient generation of spin current by the spin Nernst effect.

[1] Department of Materials Science and Engineering and KI for Nanocentury, KAIST, Daejeon 34141, Korea. [2] Department of Materials Science and Engineering, Graduate School of Energy Science Technology, Chungnam National University, Daejeon 34134, Korea. [3] Department of Materials Science and Engineering, Korea University, Seoul 02841, Korea. [4] KU-KIST Graduate School of Converging Science and Technology, Korea University, Seoul 02841, Korea. Correspondence and requests for materials should be addressed to B.-G.P. (email: bgpark@kaist.ac.kr)

A central theme of spintronics field is the electrical generation of a spin current as the spin current allows for an efficient magnetization switching and a high speed domain wall motion in magnetic nanostructures[1–6]. In ferromagnet (FM)/non-magnetic heavy metal (HM) bilayers, a longitudinal charge current creates a transverse spin current via spin Hall effect (SHE)[7, 8]. The spin current induces spin accumulation at the FM/HM interface, which exerts a torque on the FM and controls the magnetization direction[1, 2]. On the other hand, the spin current is partially reflected from the FM/HM interface depending on its spin orientation with respect to the magnetization direction of the FM layer. This reflected spin current is then converted to a charge current via inverse SHE (ISHE), resulting in the variation of the longitudinal resistance of the FM/ HM bilayers, i.e., spin Hall magnetoresistance (SMR)[9–11]. As the SMR originates from the SHE-induced spin current and the ISHE of the reflected spin current, its magnitude depends on the square of the spin Hall angle ($\theta_{\mathrm{SH}}$), charge-to-spin conversion efficiency.

Spin current is also generated by a temperature gradient, for instance, the spin (-dependent) Seebeck effect in FM/non-magnetic bilayer structures where thermally induced spin current is injected from the FM into the non-magnetic layer[12–17]. Theories have predicted that a pure spin current is thermally generated in non-magnetic materials by their spin–orbit coupling effects[18–21], a thermal analog to the SHE, i.e., spin Nernst effect (SNE) (Fig. 1a). However, there has been no experimental observation yet on the thermally induced pure spin current or SNE.

In this work, we demonstrate the SNE by investigating the Hall resistance variation of the FM/HM bilayers under a temperature gradient. Similar to the SMR originating from combined effects of SHE (charge-to-spin conversion) and ISHE (spin-to-charge conversion), the SNE could also cause a resistance variation of the bilayer. This thermally induced magnetoresistance in a FM/HM bilayer, which can be called spin Nernst magnetoresistance (SNMR), originates from combined effects of two processes: (i) SNE-induced spin current in HM, of which efficiency is described by spin Nernst angle ($\theta_{\mathrm{SN}}$), heat-to-spin conversion efficiency, and (ii) subsequent reflection of a spin current at the FM/HM interface and conversion to a charge current via ISHE, of which efficiency is described by $\theta_{\mathrm{SH}}$ (Fig. 1b). As a result, the magnitude of SNMR is determined by the product of $\theta_{\mathrm{SN}}$ and $\theta_{\mathrm{SH}}$. In analogous to a modification of the planar Hall effect signal (i.e., transverse SMR) by the SHE[11, 22], the SNE modifies the planar Nernst effect signal (i.e., transverse SNMR). Therefore, a systematic investigation of transverse SNMR in FM/HM bilayers allows us to identify the SNE, which we have done in this work.

We find that $\theta_{\mathrm{SN}}$ has a comparable magnitude to $\theta_{\mathrm{SH}}$ for W and Pt, suggesting that the SNE can efficiently create a spin current as much as the SHE can do.

## Results

**Spin Nernst magnetoresistance model.** The SNMR in a FM/HM bilayer can be described by replacing SHE-induced spin current with thermal pure spin current, $J_{s,T} = -\theta_{\mathrm{SN}}\sigma_{\mathrm{HM}}S_{\mathrm{HM}}\frac{\partial T}{\partial x}$, in the SMR model[9–11], where $\sigma_{\mathrm{HM}}$ and $S_{\mathrm{HM}}$ are the electrical conductivity and the Seebeck coefficient of the HM, respectively. The longitudinal ($\Delta V_{xx}$) and transverse ($\Delta V_{xy}$) thermoelectric voltages caused by the longitudinal and transverse SNMRs are respectively expressed as,

$$\frac{\Delta V_{xx}}{L_{\mathrm{V}}} = -\left[S_0 + \Delta S_1 + \Delta S_2\left(1 - m_y^2\right)\right]\frac{\Delta T_x}{L_{\mathrm{T}}}, \tag{1}$$

$$\frac{\Delta V_{xy}}{L_{\mathrm{V}}} = -\left[\Delta S_2 m_x m_y + \Delta S_3 m_z\right]\frac{\Delta T_x}{L_{\mathrm{T}}}, \tag{2}$$

where $L_{\mathrm{T(V)}}$ is the effective length for temperature gradient (thermal voltage generation), $m_x$, $m_y$, and $m_z$ are the $x$, $y$, and $z$ component of the magnetization, respectively, $\Delta T_x$ is the temperature difference along the $x$-axis induced from localized thermal excitation, and

$$\Delta S_1 \equiv -\chi_{\mathrm{HM}}\theta_{\mathrm{SH}}\theta_{\mathrm{SN}}S_{\mathrm{HM}}\frac{2\lambda}{d_{\mathrm{HM}}}\tanh\left(\frac{d_{\mathrm{HM}}}{2\lambda}\right), \tag{3}$$

$$\Delta S_2 \equiv \chi_{\mathrm{HM}}\theta_{\mathrm{SH}}\theta_{\mathrm{SN}}S_{\mathrm{HM}}\mathrm{Re}\frac{\lambda}{d_{\mathrm{HM}}}\frac{2\lambda G\tanh^2\left(\frac{d_{\mathrm{HM}}}{2\lambda}\right)}{\sigma_{\mathrm{HM}} + 2\lambda G\coth\left(\frac{d_{\mathrm{HM}}}{\lambda}\right)}, \tag{4}$$

$$\Delta S_3 \equiv -\chi_{\mathrm{HM}}\theta_{\mathrm{SH}}\theta_{\mathrm{SN}}S_{\mathrm{HM}}\mathrm{Im}\frac{\lambda}{d_{\mathrm{HM}}}\frac{2\lambda G\tanh^2\left(\frac{d_{\mathrm{HM}}}{2\lambda}\right)}{\sigma_{\mathrm{HM}} + 2\lambda G\coth\left(\frac{d_{\mathrm{HM}}}{\lambda}\right)}, \tag{5}$$

and $S_0$ is the ordinary Seebeck coefficient in the bilayer structure. Here, $\Delta S_1$, $\Delta S_2$, and $\Delta S_3$ are additional Seebeck coefficients induced by SNE, where $d_{\mathrm{HM}}$ and $\lambda$ are the thickness and spin diffusion length of the HM, respectively, and $G$ is the spin mixing conductance of the FM/HM interface. Note that SNE in FM layer and inverse SNE in HM are ignored and the shunting effect of FM layer is taken into consideration using a geometric factor, $\chi_{\mathrm{HM}} = ((\sigma_{\mathrm{HM}}d_{\mathrm{HM}})/(\sigma_{\mathrm{HM}}d_{\mathrm{HM}} + \sigma_{\mathrm{FM}}d_{\mathrm{FM}}))$, where $\sigma_{\mathrm{FM}}$ and $d_{\mathrm{FM}}$ are the electrical conductivity and thickness of the FM layer, respectively. The $\Delta V_{xx}$ depends on the magnetization direction relative to the spin orientation ($y$) of SNE-induced spin current, and it is thus proportional to $m_y^2$ while its magnitude is

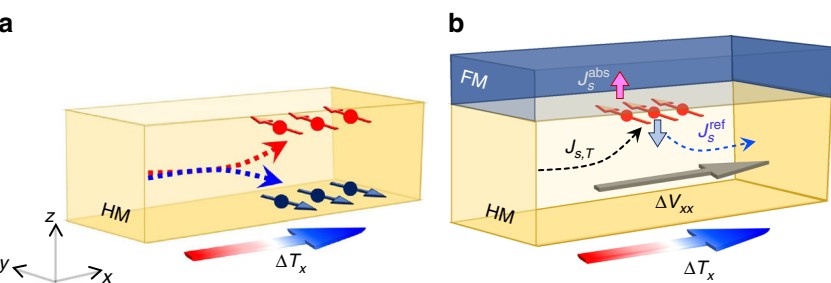

**Fig. 1** Schematics for spin Nernst effect and spin Nernst magnetoresistance. **a** Spin Nernst effect (SNE), where the temperature gradient in $x$-direction generates a spin current in $z$-direction with the spin orientation in $y$-direction. **b** Spin Nernst magnetoresistance (SNMR) in FM/HM bilayer where a spin current induced in HM by a temperature gradient in $x$-direction ($J_{s,T}$) partially reflected at the FM/HM interface depending on its spin orientation with respect to the magnetization direction of the FM layer, resulting in the modification of the longitudinal ($\Delta V_{xx}$) and transverse ($\Delta V_{xy}$) thermoelectric voltages of the bilayer. $J_s^{\mathrm{abs}}$($J_s^{\mathrm{ref}}$) is the absorbed (reflected) spin current at the FM/HM interface

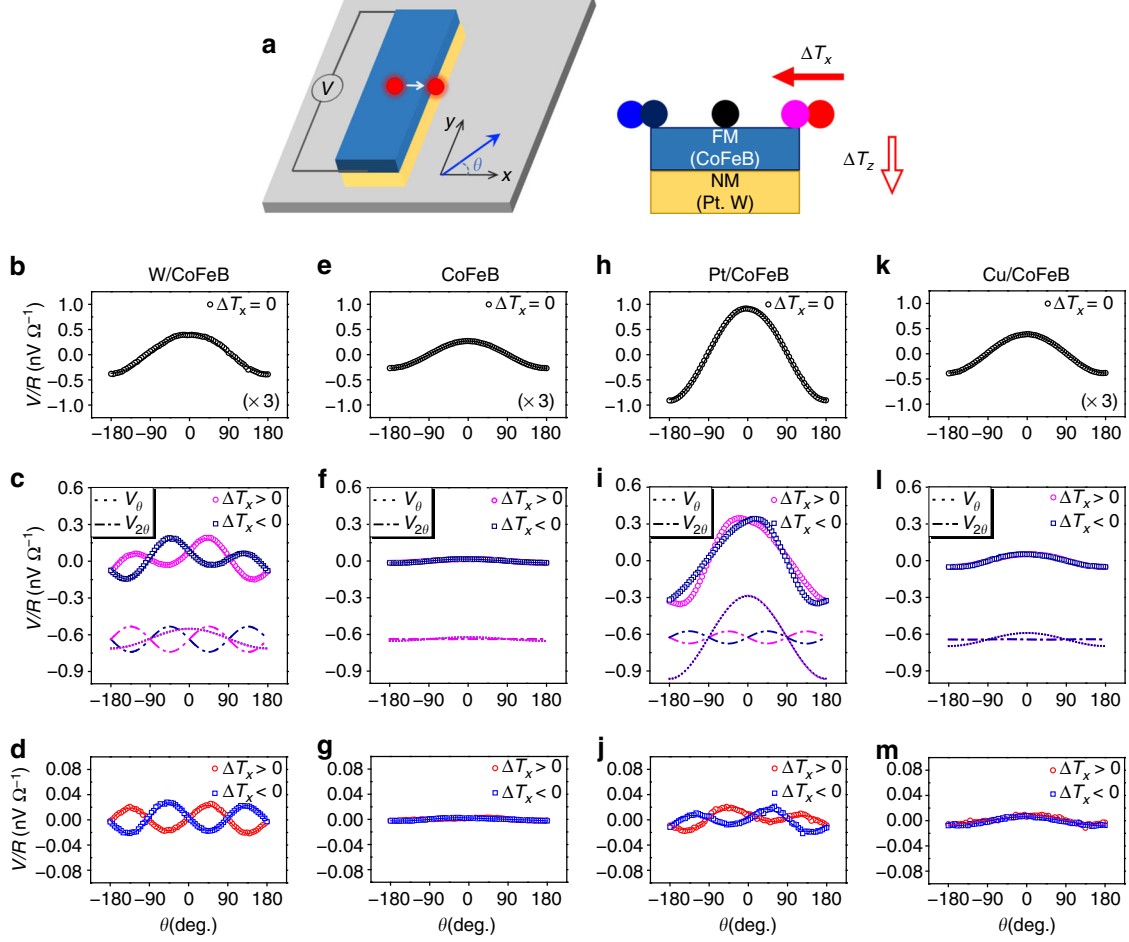

**Fig. 2** Transverse spin Nernst magnetoresistance originating from SNE in various layer structures. **a** Schematics of measurement under different laser position on bar-shaped structure. The $x$–$z$ plane view indicates the laser positions along the $x$ direction. Each color of circle represents the laser position. **b**–**m** Thermoelectric Hall signals for W(3 nm)/CoFeB(2 nm) (**b**–**d**), CoFeB(2 nm) (**e**–**g**), Pt(3 nm)/CoFeB(2 nm) (**h**–**j**), and Cu(3 nm)/CoFeB (2 nm) structures (**k**–**m**) for different laser locations, at the center ($x \sim 0$ μm, **b**, **e**, **h**, **k**), edge ($x \sim 5$ μm, **c**, **f**, **i**, **l**), and outside of the structure ($x \sim 10$ μm, **d**, **g**, **j**, **m**) for each sample, which are normalized by sample resistance. Dotted and dash-dotted lines (**c**, **f**, **i**, **l**) show the decomposition of two angle-dependent signals of $\cos\theta$ and $\sin 2\theta$. The symbol color denotes the laser position as illustrated in schematics of Fig. 2a

determined by the $\Delta S_2$. As our samples have in-plane magnetization ($m_z \approx 0$), the $\Delta V_{xy}$ ($\propto m_x m_y$) has the same magnitude ($\Delta S_2$) as that of the $\Delta V_{xx}$, so that the investigation of $\Delta V_{xy}$ corresponding to the transverse SNMR (or planar Nernst effect (PNE) signal) allows us to explore the SNE. Note that the sign of the SNMR is determined by the sign of the product of $\theta_{SH}$, $\theta_{SN}$, and $S_{HM}$ of the HM, which is distinct from the fact that the sign of the SMR is independent of the sign of $\theta_{SH}$.

**Transverse spin Nernst magnetoresistance in W/CoFeB.** We first examine the transverse SNMR in W(3 nm)/Co$_{32}$Fe$_{48}$B$_{20}$(-CoFeB, 2 nm) sample, in which a thermal gradient is generated by a focused laser (55 mW) of ~5 μm diameter. Figure 2a schematically illustrates the experiment setup where thermoelectric Hall voltage is measured as a function of in-plane magnetic field angle $\theta$ with respect to the $x$-axis under a temperature gradient. The magnetization is aligned parallel to the applied magnetic field of 100 mT. Depending on the laser position in the sample structure, a vertical ($\Delta T_z$) and/or lateral ($\Delta T_x$) temperature differences in the sample are created accordingly (Supplementary Note 1). Upon illumination with a laser spot at the center of the sample

(Fig. 2a, b), generating only $\Delta T_z$ while $\Delta T_x$ cancels out, the thermoelectric signal of W/CoFeB sample shows a clear $\cos\theta$ dependence ($\propto m_x$); the largest value (zero) for $\theta = 0$ ($\theta = \pm 90$), where the magnetization is aligned to the $x$-axis ($y$-axis). This reveals that the signal originates from the longitudinal spin Seebeck effect and anomalous Nernst effect[23]. On the other hand, as the laser spot moves toward the edge of the sample, the laser illumination generates non-zero $\Delta T_x$ and as a result, an additional angle-dependent thermoelectric Hall signal appears, which is proportional to $m_x m_y$ or $\sin 2\theta$. The $\sin 2\theta$ signal reverses its sign upon the change in the direction of $\Delta T_x$ while the $\cos\theta$ signal remains the same sign, which is demonstrated in the Fig. 2c where two angle-dependent signals are decomposed. The $\sin 2\theta$ signal eventually dominates the total signal when the laser spot moves further away, where $\Delta T_z$ induced in the sample is negligible. (see the Fig. 2d), confirming that it originates from $\Delta T_x$. Note that the thermoelectric signals are almost independent of the magnetic field when it is large enough to saturate the magnetization (Supplementary Note 2), and the laser position along the $y$-axis (Supplementary Note 3). The latter is due to the local excitation by the laser heating (diameter ~5 μm) in the elongated

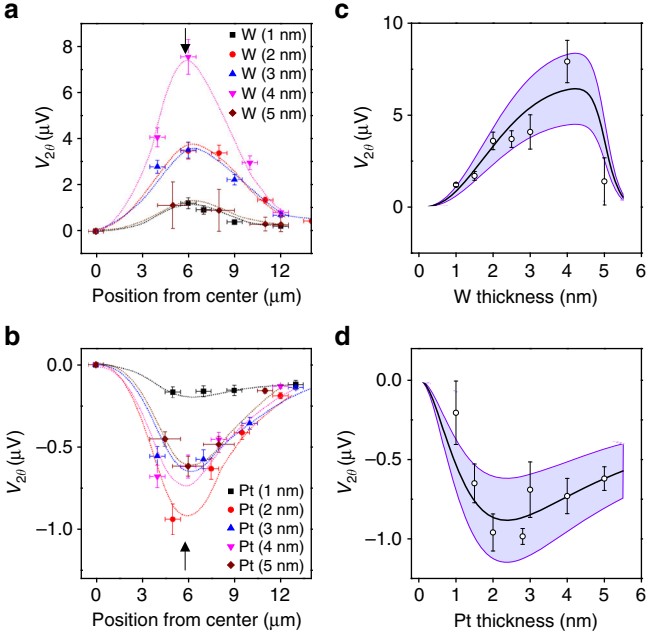

**Fig. 3** Thickness dependence of transverse SNMR in W/CoFeB and Pt/CoFeB structures. **a**, **b** Laser-position-dependent thermoelectric signal $V_{2\theta}$ for W/CoFeB structure (**a**) and Pt/CoFeB structure (**b**) with different HM thicknesses ranging from 1 to 5 nm. Black arrow represents position of edge illumination. **c**, **d** HM thickness dependence of the $V_{2\theta}$ for edge illumination for W/CoFeB structure (**c**) and Pt/CoFeB structure (**d**). The white circles represent experimental data and solid lines represent best fitted curves, while purple band indicates error ranges of extracted values, which originated from uncertainties of $S_{HM}$, $G$, and $\theta_{SH}$. Error bars represent the range of the measured values, resulting from laser-position uncertainty

sample structure: 10 μm × 1 mm in which the $\Delta T_y$ between the two ends of the sample is not significantly generated by the laser illumination.

The signal induced by $\Delta T_x$ ($\propto m_x m_y$) has the same symmetry as PNE, but the magnitude is noticeably large in the W/CoFeB sample as compared to that in the control sample of a single CoFeB (2 nm) layer (see the decomposed dotted lines in the Fig. 2f). As the $\Delta T_x$ in a single CoFeB is comparable to that of W/CoFeB layer (Supplementary Note 1), the large enhancement in PNE indicates that there is a significant contribution from the W layer to the sin 2θ thermoelectric signal, which we attribute to the consequence of transverse SNMR caused by a thermal generation of pure spin current in W and subsequent reflection of the spin current at the W/CoFeB interface depending on the magnetization direction. Note that the enhanced PNE signal is due to the transverse component of the SNMR as the planar Hall effect in the same W/CoFeB sample is strongly modified by the transverse SMR (Supplementary Note 4). As the transverse SNMR depends on SNE-induced spin current and its conversion into charge voltage via the ISHE, the sign of the transverse SNMR and equivalently the sign of the PNE corresponding to the SNE are determined by the product of the $\theta_{SH}$ and $\theta_{SN}$. For W, it is known as $\theta_{SH} < 0$[11, 22, 24] and $S_{HM} > 0$[25], thus the positive transverse SNMR for $\Delta T_x > 0$ indicates that a positive $\theta_{SN}$ for W, which is the opposite sign to its $\theta_{SH}$. We note that this sign difference is not impossible because $\theta_{SH}$ is determined by the density of states at the Fermi energy while $\theta_{SN}$ is determined by the energy derivative of density of states[20].

**Material dependence of spin Nernst magnetoresistance**. We also investigate the transverse SNMR for different non-magnetic materials such as Pt and Cu. Note that Pt has a positive $\theta_{SH}$[9, 12, 24, 26], the opposite sign to that of W, while Cu has a negligible $\theta_{SH}$[9, 24]. Figure 2h, k show that under the central heating, both samples exhibit cos θ angular dependence as the W/CoFeB or CoFeB sample does (see the Fig. 2b, e). When a sizable $\Delta T_x$ is applied, on the other hand, the sin 2θ thermoelectric signal exhibits a strong material dependence; an opposite sign for the Pt/CoFeB sample (Fig. 2i, j) and negligibly small for the Cu/CoFeB sample (Fig. 2l, m) as compared to that of the W/CoFeB sample. As the same thickness of CoFeB is used and a similar $\Delta T_x$ is induced for all samples (Supplementary Note 1), these results again confirm that the sin 2θ thermoelectric signal is dominated by the SNE-induced spin current in HM through its spin–orbit coupling effects.

**Estimation of spin Nernst angle**. We next estimate the heat-to-spin conversion coefficient $\theta_{SN}$ using the HM layer thickness dependence of the transverse SNMR in HM/CoFeB samples. We note that the accuracy of this estimation substantially depends on the accuracy of $\Delta T_x$ and $\Delta T_z$. As it is hard to experimentally determine $\Delta T_x$ and $\Delta T_z$, we estimate the temperature distribution of the sample under the laser illumination by solving the heat transfer module of the COMSOL software (Supplementary Note 1) and a control sample (Supplementary Fig. 5). As a result, we do not argue that our estimation of $\theta_{SN}$ is quantitatively accurate, but we believe that it is still meaningful to estimate $\theta_{SN}$ even approximately.

We performed the same measurement shown in Fig. 2 while varying the laser positions from the center to the edge of the samples, and then separated the cos θ and sin 2θ components ($V_\theta$, $V_{2\theta}$). The latter corresponds to the transverse SNMR which is summarized in Fig. 3a, b for W/CoFeB and Pt/CoFeB samples, respectively (see Supplementary Note 5 for more details). The $V_{2\theta}$ shows the peak values when the laser is located at the edge of the sample (x ~ 5 μm), where $\Delta T_x$ is maximized. Figure 3c show the $V_{2\theta}$ of W/CoFeB samples for the edge illumination as a function of W thickness, demonstrating that the $V_{2\theta}$ becomes the largest at 4 nm of W and decreases with a further increase in W thickness. This is the same trend as the W thickness dependence of the SMR in similar W/CoFeB structures (ref. [22] and Supplementary Note 4), indicating that the spin transport in W dominantly governs the transverse SNMR of our samples. A similar thickness dependence of the transverse SNMR is also observed for the Pt/CoFeB samples, which is shown in Fig. 3d. In order to estimate $\theta_{SN}$, we fit the thickness dependence of the transverse SNMR to Eq. (2) using material parameters (Table 1) and the calculated $\Delta T_x$ that is obtained to be ~24 K for W/CoFeB and Pt/CoFeB samples, and ~17 K for CoFeB sample when the laser of 55 mW illuminates at the edge of the sample (Supplementary Fig. 4). Note that the variation of resistivity in W with its thickness has been taken into account (Supplementary Note 6). From the fitting, we obtained $\theta_{SN}$ values of 0.22–0.42 for W and −0.12 to −0.24 for Pt, and λ values of $(2.0 \pm 0.1)$ nm for W, and $(1.0 \pm 0.1)$ nm for Pt. The purple bands in Fig. 3c, d indicate error ranges which possibly originates from uncertainties (±30%) of the literature values of $S_{HM}$, $G$, and $\theta_{SH}$. Note that the Seebeck coefficient of Pt is negative ($S_{HM} < 0$)[25], which is an opposite sign to that of W. This fitting result demonstrates that $\theta_{SN}$ has a comparable magnitude to $\theta_{SH}$ but has an opposite sign to $\theta_{SH}$ for both W and Pt (Table 1). The comparable magnitude between $\theta_{SN}$ and $\theta_{SH}$ implies that the SNE in HM layer can create a spin current as much as the SHE can if a thermal gradient is efficiently generated.

**Table 1 Parameters for analysis of transverse spin Nernst magnetoresistance**

| | $1/\sigma_{HM}$ ($\mu\Omega$ cm) | $1/\sigma_{FM}$ ($\mu\Omega$ cm) | $\theta_{SH}$ | $G$ ($\Omega^{-1}$ m$^{-2}$) | $S_{HM}$ ($\mu$V K$^{-1}$) | $\lambda$ (nm) | $\theta_{SN}$ |
|---|---|---|---|---|---|---|---|
| W/CoFeB | 35–125 | 320 | $-0.21$[22, 24] | $0.5$–$5 \times 10^{15}$[11, 22] | $10$[25] | $2.0 \pm 0.1$ | 0.22 to 0.42 |
| Pt/CoFeB | 30 | 320 | $0.10$[24, 26] | $0.5$–$5 \times 10^{15}$[10, 26] | $-10$[25] | $1.0 \pm 0.1$ | $-0.12$ to $-0.24$ |

## Discussion

We demonstrate the transverse SNMR in HM/FM bilayers which signifies an efficient thermal generation of spin current by SNE. Our estimation of the heat-to-spin conversion efficiency $\theta_{SN}$ of W or Pt implies that the magnitude of $\theta_{SN}$ could be comparable to that of the charge-to-spin conversion efficiency $\theta_{SH}$. This suggests that the SNE-induced spin current could create a considerable spin torque to adjacent FM layer, or thermal spin–orbit torques that can manipulate the magnetization direction of the FM as electrical spin–orbit torques do. Moreover, thermal spin–orbit torque can be combined with electrical spin–orbit torque by applying both a charge current and a thermal gradient to bilayers, which allows for the reduction in the critical current for magnetization switching. These results open up an alternative way to generate the spin current and/or to control the magnetization direction in spintronic devices.

We would like to note that while we were preparing the manuscript, we became aware that similar work has been done by other groups[27, 28].

## Methods

**Sample preparation**. All samples of W/Co$_{32}$Fe$_{48}$B$_{20}$(CoFeB), Pt/CoFeB, and CoFeB were prepared by magnetron sputtering on thermally oxidized Si substrates with a base pressure of less than $4.0 \times 10^{-6}$ Pa ($3.0 \times 10^{-8}$ Torr) at room temperature. All samples were covered by MgO (1 nm)/Ta (1 nm) capping layer to prevent oxidation. The bar-shaped structures of 10 $\mu$m $\times$ 1 mm dimension for SNMR measurement are patterned using photolithography and Ar ion milling. The resistivities are measured to be $320 \times 10^{-8}$ $\Omega$ m$^{-1}$ for CoFeB, $30 \times 10^{-8}$ $\Omega$ m$^{-1}$ for Pt, while that of W is $112 \times 10^{-8}$ $\Omega$ m$^{-1}$ when W is thinner than 4 nm and it gradually decreases with its thickness greater than 4 nm.

**Transverse spin Nernst magnetoresistance measurements**. The thermoelectric Hall voltage along the $y$-axis was measured under the temperature gradients ($\nabla T_x, \nabla T_z$) in the sample, which were generated by laser illumination of 55 mW, while rotating a magnetic field of 100 mT in the $x$–$y$ plane which is larger than in-plane anisotropy field of CoFeB layer. The measurements were repeated at each laser position varying from center to edge of the sample, which was monitored by its reflectance of the laser. All measurements were carried out at room temperature and each measurement was repeated more than three times; data are reproducible.

**Data availability**. Authors can confirm that all relevant data are included in the paper and/or its supplementary information files and data are available on request.

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

## Acknowledgements

This work was supported by the National Research Foundation of Korea (NRF-2015M3D1A1070465, 2014R1A2A1A11051344, and 2016R1A2B4012847 and 2017R1A2B2006119) and the DGIST R&D Program of the Ministry of Science, ICT and Future Planning (17-BT-02).

## Author contributions

B.-G.P. planned and supervised the study. D.-J.K. and C.-Y.J. fabricated devices. D.-J.K., C.-Y.J., J.-G.C. and J.W.L. performed spin thermoelectric and transport measurement.

S.S. and J.-R.J. simulated the temperature distribution. D.-J.K., K.-J.L. and B.-G.P. analyzed the results and wrote the manuscript.

## Additional information

**Competing interests:** The authors declare no competing financial interests.

**Change history:** A correction to this article has been published and is linked from the HTML version of this paper.

