## [Peer Review File · Nature Communications]

PEER REVIEW FILE

Reviewers' comments:

Reviewer #1 (Remarks to the Author):

Kim and colleagues report on the thermal counterpart of the spin Hall effect, pure spin current generation transverse to a temperature gradient, i.e. the spin Nernst effect. They measure the voltage drop across a micro wire made of CoFeB and CoFeB/normal metal bilayers (W, Pt and Cu) as a function of in-plane magnetization direction in the presence of vertical and lateral temperature gradients. They use a focused laser beam as a heat source shined directly on top of the wire, and depending on the laser spot location they can tune the in-plane temperature gradient transverse to the measurement axis. The in-plane and out-of-plane temperature gradients systematically generate voltages proportional to $\cos(\theta)$ and $\sin(2\theta)$, respectively, where θ is the in-plane angle of magnetization. The latter signal changes sign between W/CoFeB and Pt/CoFeB micro wires, which the authors attribute to thermally generated spin Hall effect (spin Nernst effect) modulating the transverse conductivity. They further study the W and Pt thickness dependence of the effect which support their hypothesis.

Understanding the spin-charge mechanics under the influence of a temperature gradient is a central topic in spintronics. The spin Nernst effect in particular, has attracted a recent interest since it provides an alternative way to generate pure spin currents in the absence of a charge current flow. Thus far, most studies remained at a theoretical level (ref.18-21 - manuscript), whereas a couple of experimental studies recently appeared on arXiv (ref.27,28 - manuscript), which are also acknowledged by the authors.

I believe that this manuscript shows convincing evidence of the spin Nernst effect through spin Nernst magnetoresistance. The experiments and data analysis are fair. The most convincing result is the sign change of the signal between W and Pt-based structure, which cannot be explained by other known transport effects. However I have several questions/comments to the authors as listed here:

1)The expected sign of the spin Nernst magnetoresistance rely on the product of the spin Hall angle, spin Nernst angle and Seebeck coefficient (Eq.3-5). I believe that this must be explicitly written where the equations appear and supported by illustrations for an intuitive understanding of the sign of the signal. For instance, for someone who is familiar with the spin Hall magnetoresistance, the sign change is not easily conceivable since both W and Pt based layers have the same SMR sign. The sign determination should appear more clearly throughout the

manuscript.

2) x_{HM} is introduced in Eqs.3-5 as a geometric factor. Since the spin current is generated by a temperature gradient why this factor takes into account only the electrical conductivity? Otherwise said, do the authors expect an inhomogeneous temperature gradient (along x) within the bilayer, scaled with the thermal conductivity of each layer? If yes, how is this taken into account?

3) The authors state that the SNE in CoFeB is neglected. Can they justify this choice? If it weren't neglected, what would be its possible influence on the data?

4) The measurements were performed at a fixed external field of 100 mT. I would highly recommend to perform the same measurements at different fields and separate the field-dependent signals, if any, and the magnetization-dependent signals.

5) Although the measurements are straightforward and the data presented in fig.2 are quite clear, the thickness dependence data has a large error bar and the corresponding fit is not very convincing. For Pt the authors find a spin diffusion length of 0.8 nm which is one of the lowest in literature. The estimations and the fit would be much more accurate if there were more data points (especially below 2 nm) and less scattering.

6) The simulations are described in detail in the supplementary information, however some important numbers and a concise summary must be provided in the main text. A paper at this level must be self-contained.

7) In Eq.S1 what is the physical meaning of the constant C ?

If the authors' responses are satisfactory and the manuscript is properly revised, I am inclined to believe that this work would be suitable for publication in Nature Communications.

Reviewer #2 (Remarks to the Author):

Kim et al. report an interesting observation of spin Nernst effect induced magnetoresistance, so called SNEM, in the HM(Pt, W, Cu)/CoFeB bilayers. The observation absolutely boosts the spintronic community after novel findings in the spin Seebeck effect, spin Hall effect, and etc. However, my concern is the similar works had been conducted by both Meyer et al. and Sheng et al. in July last year published in arXiv. on the similar HM/ferromagnets hybrids. This work

seems did comprehensive angular dependent studies on the SNEM, compared with previous two works. However, the novelty of SNE is definitely degraded a lot and repeat of similar works prevent me accepting this work publish in a high rank journal, like Nat. Commun. It will not be fair for others. Besides this important reason, there are also some other scientific concerns stop me to accept it to publish at the present stage, as listed below.

1) In the abstract, authors claim “spin current can be also created by a temperature gradient, which is known as spin Nernst effect”. The statement is misleading. The term of thermal generated spin current strongly depends on the materials, for example, the spin Seebeck effect is also generated by thermal gradient as well. One should avoid uncertain wording. Similar wording also can be found in P.2 and P.5.

2) The major concern is the accurate determination of temperature gradient by means of laser heating. It is generally agree that heating in this matter contains lots of uncertainty and uncontrollable factors. I am not certain this is the best/effective way to generate temperature gradient. The temperature differences in the x- and z- axis can reach up to 25 K and 50 mK. The value depends on the simulation, not a direct measurement. As this is an important factor for accurate determine the theta(SNE), the better way should be carefully redesigned and provide more convincing results. I am not sure how you could measure the RT curves on Pt and W stripes and use it as a temperature-sensors, for example.

3) Again the temperature determination, if one supposedly accept the simulation, the author should also consider the thermal conductivity of the capping layers, like MgO and Ta. These may greatly affect the thermal gradients as well.

4) In p.4, are you saying the measurements is a combine of two effects: SSE and SNE? The separation of two effect is due to the fits of two components ($V(\theta)$ and $V(2\theta)$)? Any other component should be considered?

5) Have authors consider the position dependence of y-axis? The in-plane lateral thermal gradient along the y could also produce a conventional Seebeck effect in both CoFeB and HMs. How one can be convinced there is no effect?

6) The value of spin diffusion length for Pt and W is largely deviated from the known values. Some literature reported the spin diffusion length of Pt and W is in the order of 3-5 nm. The values calculated in this work are nearly one order of magnitude smaller than previous reports. Please explain.

7) I am also not very pleased with fittings in Fig.3. Four dots could fit whatever formula with a reasonable agreement.

8) Technique concerns: I like the sketch drawn in the figures, you may think a more catch-eye drawing. I struggled always on the voltage scales for all the figures. If authors compared the physical component with different samples. This scale should be uniformed.

Reviewer #3 (Remarks to the Author):

In the work ,Observation of transverse spin Nernst magnetoresistance induced by thermal spin current in ferromagnet/non-magnet bilayers‘ the authors present the observation of the so-called spin Nernst effect, the thermal driven analogue to the spin Hall effect. An in-plane temperature gradient along a W/CoFeB bilayer leads to a transversely converted spin current, which was observed via transverse ,SMR-like‘ measurements called SNMR. The measurements were compared to a CoFeB reference sample.

The authors can well-founded present the generation of a pure in-plane temperature gradient by using a focused laser beam. The detailed investigation of different laser beam positions and the analysis of SSE/ANE ($\cos(\theta)$) and PNE ($\sin(2\theta)$) contributions are necessary and well convincing. Especially the additional investigations in the supplemental material is profound. Furthermore, the theoretical considerations concerning the magnetization dependent Seebeck coefficients, interplay of spin Hall and spin Nernst effects, spin mixing conductance and shunting effects, etc., indicate a rigorous scientific work on this special topic.

The investigation of $\cos(\theta)$ and $\sin(2\theta)$ contributions were performed for different systems, i.e., CoFeB, W/CoFeB, Pt/CoFeB and Cu/CoFeB. The choice for these material combinations is very reasonable when spin Hall and spin Nernst related transport phenomena are investigated. The different sign of the spin Hall angle in W and Pt and the negligible spin Hall angle in Cu expose the real existence of these effects. The authors could show that there is a change of sign in the observed signals between W and Pt used as the heavy metal (HM) and there is no spin Hall/Nernst contribution when Cu or no HM material is used.

The next step is unavoidably a thickness dependency of the used HM material. These measurements exhibit a maximum signal for a certain W and Pt thickness, respectively. This implies to the spin diffusion length of the used HM materials and corroborates to a spin Hall/Nernst effect as the origin. A laser beam position dependency revealed to a maximum signal for a certain position which lead to a maximum in-plane temperature gradient.

Conclusively, the authors accomplished their analysis with a calculation of the spin Nernst angle by using materials parameters from the literature (spin mixing conductance, Seebeck coefficients, etc.). They found spin Nernst angles of W and Pt very similar to their spin Hall angles but with an opposite sign.

The presented work is profound and scientifically well analyzed. The spin Nernst effect was investigated by regarding all important side effects which can be obtained by using temperature gradients, i.e., planar Nernst and anomalous Nernst effect as well as spin Seebeck effect, when unintended temperature gradients are involved. For this reason a detailed temperature gradient

analysis was performed which is necessary in the field of spin caloric transport phenomena. I would like to recommend this manuscript for publication in Nature Communications. However, there are a few points I would like to mention and I hope they will be commented within a minor revision.

1. First of all, I was wondering how the laser spot was aligned in the y-direction. The alignment in the x-direction was well investigated and the contributions of $\cos(\theta)$ and $\sin(2\theta)$ lead to the temperature gradients in x- and z-direction. Please, can you comment on the influence of a temperature gradient along the y-direction when the laser spot is misaligned in that direction?

2. The interplay between SNE and ISHE is mentioned and leads to the SNMR effect. What about an inverse SNE effect? Can you comment on this?

3. In the supplemental materials a detailed PHE/AMR/SMR for one sample was performed. Could you see similar spin Hall related dependencies which confirm the observed spin Nernst related effects? SMR, etc.

4. You mentioned the benefits of the SNE for spin torque switching compared to electrical switching given by the SHE. Can you comment on the effect magnitudes of the SNE if it is realistic for the given sample materials to reach a temperature gradient giving a spin accumulation and a related spin torque which is similar to the spin accumulation you get for a current density in the range of 10^6 to 10^7 A/cm²?

Responses to Reviewers' Comments:

Reviewers' comments:

Reviewer #1 (Remarks to the Author):

Kim and colleagues report on the thermal counterpart of the spin Hall effect, pure spin current generation transverse to a temperature gradient, i.e. the spin Nernst effect. They measure the voltage drop across a micro wire made of CoFeB and CoFeB/normal metal bilayers (W, Pt and Cu) as a function of in-plane magnetization direction in the presence of vertical and lateral temperature gradients. They use a focused laser beam as a heat source shined directly on top of the wire, and depending on the laser spot location they can tune the in-plane temperature gradient transverse to the measurement axis. The in-plane and out-of-plane temperature gradients systematically generate voltages proportional to $\cos(\theta)$ and $\sin(2\theta)$, respectively, where θ is the in-plane angle of magnetization. The latter signal changes sign between W/CoFeB and Pt/CoFeB micro wires, which the authors attribute to thermally generated spin Hall effect (spin Nernst effect) modulating the transverse conductivity. They further study the W and Pt thickness dependence of the effect which support their hypothesis.

Understanding the spin-charge mechanics under the influence of a temperature gradient is a central topic in spintronics. The spin Nernst effect in particular, has attracted a recent interest since it provides an alternative way to generate pure spin currents in the absence of a charge current flow. Thus far, most studies remained at a theoretical level (ref.18-21 - manuscript), whereas a couple of experimental studies recently appeared on arXiv (ref.27,28 - manuscript), which are also acknowledged by the authors.

I believe that this manuscript shows convincing evidence of the spin Nernst effect through spin Nernst magnetoresistance. The experiments and data analysis are fair. The most convincing result is the sign change of the signal between W and Pt-based structure, which cannot be explained by other known transport effects. However I have several questions/comments to the authors as listed here:

Response) First of all, we appreciate for reviewer's comment "*this manuscript shows convincing evidence of the spin Nernst effect through spin Nernst magnetoresistance. The*

experiments and data analysis are fair. The most convincing result is the sign change of the signal between W and Pt-based structure, which cannot be explained by other known transport effects”.

1) The expected sign of the spin Nernst magnetoresistance rely on the product of the spin Hall angle, spin Nernst angle and Seebeck coefficient (Eq.3-5). I believe that this must be explicitly written where the equations appear and supported by illustrations for an intuitive understanding of the sign of the signal. For instance, for someone who is familiar with the spin Hall magnetoresistance, the sign change is not easily conceivable since both W and Pt based layers have the same SMR sign. The sign determination should appear more clearly throughout the manuscript.

Response 1) Thank you for comment that the sign of the SNMR that depends on spin Hall angle, spin Nernst angle, and Seebeck coefficient should be clearly stated on the page where the equations is written. We agree that this will help the readers to understand the difference between SNMR and SMR, thus we included the following sentence on page 5 of the revised manuscript.

“Note that the sign of the SNMR is determined by the sign of the product of θ_{SH} , θ_{SN} , and S_{HM} of the HM, which is distinct from the fact that the sign of the SMR is independent of the sign of θ_{SH} ”

2) x_{HM} is introduced in Eqs.3-5 as a geometric factor. Since the spin current is generated by a temperature gradient why this factor takes into account only the electrical conductivity? Otherwise said, do the authors expect an inhomogeneous temperature gradient (along x) within the bilayer, scaled with the thermal conductivity of each layer? If yes, how is this taken into account?

Response 2) For the analysis of the SNMR, we derived Eqs. 3~5 by introducing thermal spin current in the SMR model [*Phys. Rev. Lett.* 110, 206601 (2013), *Phys. Rev. B* 87, 144411 (2013)] that is based on ferromagnetic insulator/heavy metal bilayer, in which the conversion of the spin current into the charge current and resultant voltage drop occurs only in heavy metal layer. However, we used a metallic ferromagnet in this study, so current shunting effect in metallic ferromagnet or geometric factor has to be taken into account. We note that what

we measure is the voltage which is determined by the product of spin-current-induced charge current and sample resistance. Therefore, regardless of the source of the spin current, our measurements for both SMR and SNMR involves the sample resistance and thus we need to take the resistivity of the ferromagnet (or geometric factor) into account when a metallic ferromagnet is used.

On the other hand, when we estimate the temperature profile using COMSOL software, the thermal conductivity (κ) of each layer including SiO₂ substrate is taken into consideration: $\kappa_W = 163$ W/mK, $\kappa_{Pt} = 72$ W/mK, and $\kappa_{CoFeB} = 10$ W/mK [<https://www.matweb.com>, <https://www.webelements.com>]. Therefore, non-uniform heat transport in the bilayer was already included in the calculated results shown in Supplementary Figs. 1-4.

Moreover, unlike the electrical transport that is dominated by the metallic bilayer, the heat transport is dominated mostly by the thick SiO₂ substrate and its thermal conductivity. Note that the thickness of SiO₂ is 100 nm, which is ~ 20 times larger than that of the W/CoFeB or Pt/CoFeB bilayer. This means that the temperature gradient along the x -direction in the bilayer is determined by the SiO₂ substrate. In order to verify the effect of thermal conductivities of metallic W, Pt, and CoFeB layers on the lateral temperature gradient, we performed additional calculations in which the thermal conductivities of all metallic layers are set to be the same: $\kappa_{Pt} = \kappa_W = \kappa_{CoFeB} = 10$ W/mK. Figure R1 demonstrates that the lateral temperature gradient is not considerably influenced by the change in the thermal conductivity of the HM layer.

3) The authors state that the SNE in CoFeB is neglected. Can they justify this choice? If it weren't neglected, what would be its possible influence on the data?

Response 3) As (inverse) spin Hall effect (SHE) was observed in a ferromagnetic layer [*Phys. Rev. Lett.* 111, 066602 (2013)], spin Nernst effect (SNE), a thermal analog to the SHE can also exist in ferromagnetic CoFeB layer. The SNE in CoFeB would lead to the overestimation of the spin Nernst angle of HM because we assume the SNMR totally originates from the SNE-induced spin current in HM. However, the SNMR signal ($V_{2\theta} \propto \sin 2\theta$) in single CoFeB: the variation of the transverse resistance induced by a lateral temperature gradient is observed to be negligible as compared to those in HM/CoFeB samples (Fig. 2b-d). This indicates that there is an insignificant contribution of SNE in CoFeB to the SNMR in the HM/CoFeB bilayer. Note that the graphs in Fig. 2 in the revised manuscript are modified by

normalizing the SNMR by longitudinal resistance for a proper comparison between the samples with different resistances (See the response to query 8 of reviewer #2).

4) The measurements were performed at a fixed external field of 100 mT. I would highly recommend to perform the same measurements at different fields and separate the field-dependent signals, if any, and the magnetization-dependent signals.

Response 4) We studied the SNMR as a function of magnetization angle with respect to the directions of temperature gradient and voltage probes. Therefore, in order to control the magnetization angle, we applied an external magnetic field of 100 mT which is larger than in-plane anisotropy field of CoFeB layer, guaranteeing the magnetization direction aligned to the magnetic field direction. We repeated the measurement done in Fig. 2b of the original manuscript [W(3 nm)/CoFeB(2 nm) sample] with different magnetic fields of 30 mT, 60 mT, and 100 mT. Figure R2 demonstrates that the thermoelectric signals are almost independent of the magnetic field when it is large enough to saturate the magnetization to field direction. This is included in the revised Supplementary Note 2.

5) Although the measurements are straightforward and the data presented in fig.2 are quite clear, the thickness dependence data has a large error bar and the corresponding fit is not very convincing. For Pt the authors find a spin diffusion length of 0.8 nm which is one of the lowest in literature. The estimations and the fit would be much more accurate if there were more data points (especially below 2 nm) and less scattering.

Response 5) We agree with the reviewer's comment that more data points are needed to convince our analysis. Therefore, we fabricated more samples with a thinner HM (Pt and W) and measured the SNMR, which is presented in Fig. R3. While there is a slight modification in the values of spin Nernst angles (θ_{SN}) and spin diffusion lengths (λ): θ_{SN} of 0.22 ~ 0.42 for W and -0.12 ~ -0.24 for Pt, and λ of (2.0 ± 0.1) nm for W, and (1.0 ± 0.1) nm for Pt, they are consistent with the results of the original manuscript, confirming the validity of our analysis of the SNMR. This is incorporated in Fig. 3 of the revised manuscript.

Moreover, we have investigated spin diffusion length of Pt and W reported in literatures (Table R1-2), where we only selected the spin diffusion lengths (λ) obtained using the ferromagnet/Pt (W) bilayer structures. There are a wide range of the λ of 1.1~4.0 nm for Pt, and 1.3~3.0 nm for W, demonstrates that our value of (1.0 ± 0.1) nm for Pt is not far from

those in the literatures. Note that the λ of Pt slightly increases to (1.0 ± 0.1) nm after more samples with a thin Pt were involved in the SNMR versus Pt thickness graph as shown in Fig. 3 of the revised manuscript.

6) The simulations are described in detail in the supplementary information, however some important numbers and a concise summary must be provided in the main text. A paper at this level must be self-contained.

Response 6) We have included the results of the temperature simulation on page 8 in the revised manuscript. “ ΔT_x is obtained to be ~ 24 K for W/CoFeB and Pt/CoFeB samples, and ~ 17 K for CoFeB sample when the laser of 55 mW illuminates at the edge of the sample where the transverse SNMR is maximized.”

7) In Eq.S1 what is the physical meaning of the constant C?

Response 7) We measured transverse SNMR as a function of magnetization angle, which contains a d.c. offset depending on the sample or measurement configuration as it is usually observed in Hall voltage measurement. Figure R4 shows raw data of the W(3 nm)/CoFeB (2 nm) sample, demonstrating the offset. The offset is simply subtracted when the SNMR is analyzed in Fig. 2 because it does not have an angular dependence that is a characteristic feature of the SNMR.

If the authors' responses are satisfactory and the manuscript is properly revised, I am inclined to believe that this work would be suitable for publication in Nature Communications.

Reviewer #2 (Remarks to the Author):

Kim et al. report an interesting observation of spin Nernst effect induced magnetoresistance, so called SNEM, in the HM(Pt, W, Cu)/CoFeB bilayers. The observation absolutely boosts the spintronic community after novel findings in the spin Seebeck effect, spin Hall effect, and etc. However, my concern is the similar works had been conducted by both Meyer et al. and Sheng et al. in July last year published in arXiv. on the similar HM/ferromagnets hybrids. This work seems did comprehensive angular dependent studies on the SNEM, compared with previous two works. However, the novelty of SNE is definitely degraded a lot and repeat of similar works prevent me accepting this work publish in a high rank journal, like Nat. Commun. It will not be fair for others. Besides this important reason, there are also some other scientific concerns stop me to accept it to publish at the present stage, as listed below.

Response) Thank you for your comments. We understand your concern on the novelty of our work. However, we can honestly state that our work is not the results from a simple repetition of those works, but rather from independent and parallel research. We found the reports in arXiv when we completed the experimental work and already started writing this manuscript. This work was a continuation of our previous studies: spin Hall magnetoresistance (SMR) in W/CoFeB/MgO [*Sci. Rep.* **5**, 14668 (2015)] and thermoelectric effect in Pt/CoFeB structures using laser heating [*Sci. Rep.* **5**, 10249 (2015)]. More importantly, our work is distinct from the arXiv reports in two points; (i) We investigated the SNMR effect in samples with various non-magnetic metals (NM) of Pt, W, and Cu, demonstrating that the magnitude and even the sign of the SNMR depends on the NM materials and their spin Hall angles (or spin Nernst angles). The sign change in the SNMR confirms that the SNMR effect is dominated by spin Nernst effect-induced spin current in NM through its spin-orbit coupling effect. (ii) We demonstrated a clear angular dependence of the SNMR signal (a large signal to noise ratio) as you pointed out. This is attributed to the laser heating system we employed in this work which induces a large temperature gradient. The large SNMR effect and its sign difference between the W and Pt-samples allow us to unambiguously demonstrate a signature of the spin Nernst effect through the SNMR. These are also acknowledged by other reviewers as shown below.

Reviewer #1: I believe that this manuscript shows convincing evidence of the spin Nernst effect through spin Nernst magnetoresistance. The experiments and data analysis are fair. The

most convincing result is the sign change of the signal between W and Pt-based structure, which cannot be explained by other known transport effects

Reviewer #3: The investigation of $\cos(\theta)$ and $\sin(2\theta)$ contributions were performed for different systems, i.e., CoFeB, W/CoFeB, Pt/CoFeB and Cu/CoFeB. The choice for these material combinations is very reasonable when spin Hall and spin Nernst related transport phenomena are investigated. The different sign of the spin Hall angle in W and Pt and the negligible spin Hall angle in Cu expose the real existence of these effects.

1) In the abstract, authors claim “spin current can be also created by a temperature gradient, which is known as spin Nernst effect”. The statement is misleading. The term of thermal generated spin current strongly depends on the materials, for example, the spin Seebeck effect is also generated by thermal gradient as well. One should avoid uncertain wording. Similar wording also can be found in P.2 and P.5.

Response 1) We appreciate for the comment on a clear statement of the thermal spin current. In order to clearly differentiate the spin current induced by spin Nernst effect (SNE) that is the main topic of our manuscript, we use the phrases of “*pure spin current can be also created by a temperature gradient*”, “*thermally-induced pure spin current*” or “*SNE-induced spin current*”, throughout the manuscript including abstract.

2) The major concern is the accurate determination of temperature gradient by means of laser heating. It is generally agree that heating in this matter contains lots of uncertainty and uncontrollable factors. I am not certain this is the best/effective way to generate temperature gradient. The temperature differences in the x- and z- axis can reach up to 25 K and 50 mK. The value depends on the simulation, not a direct measurement. As this is an important factor for accurate determine the θ (SNE), the better way should be carefully redesigned and provide more convincing results. I am not sure how you could measure the RT curves on Pt and W stripes and use it as a temperature-sensors, for example.

Response 2) We fully agree with the reviewer’s comments that the accurate determination of the temperature difference is important to extract the spin Nernst angle, which is however difficult to measure experimentally for laser heating. As we already mentioned in the original manuscript, we do not claim the precise estimation of the spin Nernst angle because of the

ambiguity in real temperature gradient and believe that it is still meaningful to estimate the spin Nernst angle using the temperature gradient calculated by COMSOL simulation that is widely used for temperature analyses.

Here we show that the COMSOL simulation result is reasonable by performing an additional measurement. First of all, we note that a typical RT curve measurement does not allow us to estimate the temperature gradient caused by the laser illumination by the following reason. The laser illumination thermally excites a local area where sample temperature is efficiently increased; ΔT_x in our sample is estimated to be ~ 24 K, guaranteeing a large thermoelectric signal (\sim a few μ V). However, the local heating makes it difficult to measure the sample temperature by measuring the resistance because the resistance increase of the illuminated area (\sim a few micron range) does not markedly contribute to the total resistance of the sample (\sim mm size).

In order to resolve the size difference issue between the sample and the laser spot, we fabricated a Pt/CoFeB sample with a narrow wire structure as illustrated in Fig. R5a. Note that in this new sample, the resistance is dominated by the narrowest wire region of which width is even smaller than the laser spot size. We first measured the resistance of the sample as a function of the temperature in physical property measurement system (PPMS, Quantum Design), showing a linear relation between the resistance and the sample temperature (Fig. R5b). Then, we measured the variation of the resistance with a laser illumination. When the laser (55 mW) is on the wire, the resistance is increased from 1202 Ω to 1232 Ω , corresponding to the increase in temperature from 296 K to 324 K. On the other hand, when the laser is moved away from the wire of ~ 10 μ m, the resistance is 1213 Ω , which corresponds to the temperature of 307 K. As a result, ΔT_x in the sample of 10 μ m width is estimated to be 17 K, which is comparable to that obtained by simulation. Because of the non-identical sample structure, this experiment cannot determine the temperature profile in the real sample of Fig. 3; however, this confirms that the laser illumination can induce a temperature difference in the sample with a similar order of magnitude as the calculated value. This experimental result is incorporated in Supplementary Fig. 5 of Supplementary Note 1.

3) Again the temperature determination, if one supposedly accept the simulation, the author should also consider the thermal conductivity of the capping layers, like MgO and Ta. These may greatly affect the thermal gradients as well.

Response 3) We appreciate the comment on the effect of the capping on the thermal gradient, which we have not considered in the original manuscript. We performed the COMSOL simulation again by including the capping layer of MgO(1 nm)/Ta(1 nm) as shown in Fig. R6. The results show that the thin capping layer does not significantly alter the overall temperature profile, but reducing the ΔT_x by ~ 1 K and ~ 0.5 K for W/CoFeB and Pt/CoFeB samples, respectively. This minor effect of the thin capping layer is expected because the thermal conduction in the lateral direction is dominated mostly by thick SiO₂ substrate (See the response to the query 2 of reviewer #1). The new calculation results with capping layer are incorporated in Supplementary Fig. 4 of Supplementary Note 1 and utilized for the extraction of the spin Nernst angle.

4) In p.4, are you saying the measurements is a combine of two effects: SSE and SNE? The separation of two effect is due to the fits of two components ($V(\theta)$ and $V(2\theta)$)? Any other component should be considered?

Response 4) When the laser is illuminated at the centre of the sample (only ΔT_z), the thermoelectric signal shows a $\cos\theta$ dependence ($\propto m_x$) only. On the other hand, as the laser spot moves toward the edge of the sample (non-zero ΔT_x), an additional angle-dependent thermoelectric signal appears, which is proportional to $m_x m_y$ or $\sin 2\theta$. As shown in Fig. 2b and Supplementary Note 5, all thermoelectric signals can be decomposed to the two angular dependent V_θ ($\propto \cos\theta$) and $V_{2\theta}$ ($\propto \sin 2\theta$) without any other angle-dependent component.

5) Have authors consider the position dependence of y-axis? The in-plane lateral thermal gradient along the y could also produce a conventional Seebeck effect in both CoFeB and HMs. How one can be convinced there is no effect?

Response 5) We have performed the same measurement as that in Fig. 2b with varying the position along the y-axis (Fig. R7). We find that the transverse thermoelectric signals are almost identical for the measurements with different y-positions (Fig. R7b-d), indicating that there is no considerable effect of the conventional Seebeck effect in our measurement configuration. This is attributed to the local excitation by the laser heating (diameter ~ 5 μm) in the elongated sample structure: 10 $\mu\text{m} \times 1$ mm in which the ΔT_y between the two ends of

the sample is not significantly generated by the laser illumination. This confirms the SNMR signal in our sample is mostly dominated by the ΔT_x . The SNMR results with different y-positions are newly included in Supplementary Note 3.

6) The value of spin diffusion length for Pt and W is largely deviated from the known values. Some literature reported the spin diffusion length of Pt and W is in the order of 3-5 nm. The values calculated in this work are nearly one order of magnitude smaller than previous reports. Please explain.

Response 6) We have investigated spin diffusion length of Pt and W reported in literatures (Table R1-2), where we only selected the spin diffusion lengths (λ) obtained using the ferromagnet/Pt (W) bilayer structures. There are a wide range of the λ of 1.1~4.0 nm for Pt, and 1.3~3.0 nm for W, demonstrates that our value of (1.0 ± 0.1) nm for Pt is not far from those in the literatures. Note that the λ of Pt slightly increases to (1.0 ± 0.1) nm after more samples with a thin Pt were involved in the SNMR versus Pt thickness graph as shown in Fig. 3 of the revised manuscript.

7) I am also not very pleased with fittings in Fig.3. Four dots could fit whatever formula with a reasonable agreement.

Response 7) We agree with the reviewer's comment that more data points are needed to convince our analysis. Therefore, we fabricated more samples with a thinner HM (Pt and W) and measured the SNMR, which is presented in Fig. R3. While there is a slight modification in the values of spin Nernst angles (θ_{SN}) and spin diffusion lengths (λ): θ_{SN} of 0.22 ~ 0.42 for W and -0.12 ~ -0.24 for Pt, and λ of (2.0 ± 0.1) nm for W, and (1.0 ± 0.1) nm for Pt, they are consistent with the results of the original manuscript, confirming the validity of our analysis of the SNMR. This is incorporated in Fig. 3 of the revised manuscript.

8) Technique concerns: I like the sketch drawn in the figures, you may think a more catch-eye drawing. I struggled always on the voltage scales for all the figures. If authors compared the physical component with different samples. This scale should be uniformed.

Response 8) Thank you for your comment. We showed the thermoelectric voltages of

different samples with non-identical scales to demonstrate their angular dependence in Fig. 2, which makes it difficult to compare their magnitudes between samples. On the other hand, if we use the same scale for all samples, it can also make a trouble to read some graphs. Therefore, in the revised manuscript, we normalized the voltage signals by their resistances, which allows us to use the same scale and to easily compare the signals among the samples (Fig. R8 and Fig. 2 of the main text). Note that the normalization by resistance is often used to compare the thermal (or ferromagnetic resonance) spin pumping signals between different samples [*Phys. Rev. Lett.* **112**, 106602 (2014), *Adv. Funct. Mater.* 26 5507 (2016)].

Reviewer #3 (Remarks to the Author):

In the work 'Observation of transverse spin Nernst magnetoresistance induced by thermal spin current in ferromagnet/non-magnet bilayers' the authors present the observation of the so-called spin Nernst effect, the thermal driven analogue to the spin Hall effect. An in-plane temperature gradient along a W/CoFeB bilayer leads to a transversely converted spin current, which was observed via transverse, SMR-like' measurements called SNMR. The measurements were compared to a CoFeB reference sample.

The authors can well-founded present the generation of a pure in-plane temperature gradient by using a focused laser beam. The detailed investigation of different laser beam positions and the analysis of SSE/ANE ($\cos(\theta)$) and PNE ($\sin(2\theta)$) contributions are necessary and well convincing. Especially the additional investigations in the supplemental material is profound. Furthermore, the theoretical considerations concerning the magnetization dependent Seebeck coefficients, interplay of spin Hall and spin Nernst effects, spin mixing conductance and shunting effects, etc., indicate a rigorous scientific work on this special topic.

The investigation of $\cos(\theta)$ and $\sin(2\theta)$ contributions were performed for different systems, i.e., CoFeB, W/CoFeB, Pt/CoFeB and Cu/CoFeB. The choice for these material combinations is very reasonable when spin Hall and spin Nernst related transport phenomena are investigated. The different sign of the spin Hall angle in W and Pt and the negligible spin Hall angle in Cu expose the real existence of these effects. The authors could show that there is a change of sign in the observed signals between W and Pt used as the heavy metal (HM) and there is no spin Hall/Nernst contribution when Cu or no HM material is used.

The next step is unavoidably a thickness dependency of the used HM material. These measurements exhibit a maximum signal for a certain W and Pt thickness, respectively. This implies to the spin diffusion length of the used HM materials and corroborates to a spin Hall/Nernst effect as the origin. A laser beam position dependency revealed to a maximum signal for a certain position which lead to a maximum in-plane temperature gradient.

Conclusively, the authors accomplished their analysis with a calculation of the spin Nernst

angle by using materials parameters from the literature (spin mixing conductance, Seebeck coefficients, etc.). They found spin Nernst angles of W and Pt very similar to their spin Hall angles but with an opposite sign.

The presented work is profound and scientifically well analyzed. The spin Nernst effect was investigated by regarding all important side effects which can be obtained by using temperature gradients, i.e., planar Nernst and anomalous Nernst effect as well as spin Seebeck effect, when unintended temperature gradients are involved. For this reason a detailed temperature gradient analysis was performed which is necessary in the field of spin caloric transport phenomena.

I would like to recommend this manuscript for publication in Nature Communications. However, there are a few points I would like to mention and I hope they will be commented within a minor revision.

Response) First of all, we appreciate for reviewer's comment "*The presented work is profound and scientifically well analyzed*" and recommendation for the publication.

1. First of all, I was wondering how the laser spot was aligned in the y-direction. The alignment in the x-direction was well investigated and the contributions of $\cos(\theta)$ and $\sin(2\theta)$ lead to the temperature gradients in x- and z-direction. Please, can you comment on the influence of a temperature gradient along the y-direction when the laser spot is misaligned in that direction?

Response 1) We have performed the same measurement as that in Fig. 2b with varying position along the y-axis (Fig. R7). We find that the transverse thermoelectric signals are almost identical for the measurements with different y-positions, indicating that there is no considerable effect of the conventional Seebeck effect in our measurement configuration. This is attributed to the local excitation by the laser heating (diameter $\sim 5 \mu\text{m}$) in the elongated sample structure: $10 \mu\text{m} \times 1 \text{mm}$ in which the ΔT_y between the two ends of the sample is not significantly generated by the laser illumination. This confirms the SNMR signal in our sample is mostly dominated by the ΔT_x . The SNMR results with different y-positions are newly included in Supplementary Note 3.

2. The interplay between SNE and ISHE is mentioned and leads to the SNMR effect. What about an inverse SNE effect? Can you comment on this?

Response 2) We appreciate for the comment on the inverse SNE, which we have not considered in the original manuscript. If there is SNE, inverse SNE should exist as well. The question is how large portion of spin current converts to heat current, which is not involved in ISHE and thus SNMR effect. Therefore, for the accurate determination of the spin Nernst angle, the temperature variation induced by the ISNE-induced heat current has to be considered, which would be extremely difficult to estimate experimentally though. This implies that the spin Nernst angle obtained by the SNMR can be underestimated; however, the consideration of the ISNE is beyond the scope of the current work: experimental demonstration of the evidence of spin Nernst effect. We believe this work will motivate experimental and theoretical investigations on SNE as well as ISNE.

3. In the supplemental materials a detailed PHE/AMR/SMR for one sample was performed. Could you see similar spin Hall related dependencies which confirm the observed spin Nernst related effects? SMR, etc.

Response 3) We only showed the transport measurement (SMR) of one representative sample in the original manuscript as the thickness dependence of the SMR in a W/CoFeB/MgO sample was published in our previous report [*Sci. Rep.* **5**, 14668 (2015)]. Nevertheless, we performed additional transport measurements with samples with different W thicknesses (Fig. R9-R11), confirming that the SMR is the dominating magnetoresistance for all samples. Moreover, W thickness dependence of the SMR is consistent with that of the SNMR (Fig. R12). This is included in the revised Supplementary Note 4.

4. You mentioned the benefits of the SNE for spin torque switching compared to electrical switching given by the SHE. Can you comment on the effect magnitudes of the SNE if it is realistic for the given sample materials to reach a temperature gradient giving a spin accumulation and a related spin torque which is similar to the spin accumulation you get for a current density in the range of 10^6 to 10^7 A/cm²?

Response 4) When we estimate the SNE-induced spin current in our sample, using

$J_{s,T} = -\theta_{SN}\sigma_{HM}S_{HM}\frac{\partial T}{\partial x}$, where $\theta_{SN,W}=0.3$, $\sigma_W = 1/\Omega\mu m$, $S_W = 10\mu V/K$, $\frac{\partial T}{\partial x} = 24K/5\mu m$, its magnitude is about 0.7% of that induced by spin Hall effect with an electric current density of $10^6 A/cm^2$ and $\theta_{SH,W} = 0.2$. It shows that the SNE-induced spin current is not significant in the current experimental condition. [REDACTED]

[REDACTED]

[REDACTED]

[REDACTED]

Table R1. Spin diffusion length of Pt

Pt (Material structure)	Spin diffusion length	Methods	Reference
Co/Pt	1.1 nm	Spin Hall magnetoresistance	APL 107 192405 (2015)
NiFe/Pt	1.2 nm	FMR spin pumping	APL 103 242414 (2013)
NiFe /Pt	1.3 nm	FMR spin pumping	PRL 111 217602 (2013)
CoFe/Pt	1.4 nm	FMR spin pumping	PRB 92 064426 (2015)
NiFe /Pt	1.4 nm	FMR spin pumping	PRB 89 060407(R) (2014)
YIG/Pt	1.5 nm	Spin Hall magnetoresistance	PRB 87 184421 (2013)
NiFe (YIG)/Pt	1.5 nm	FMR spin pumping	PRL 110 217602 (2013)
NiFe/Pt	2.0 nm	Spin absorption	PRB 91 024402 (2015)
CoFeB/Pt	3.0 nm	Spin Seebeck effect	Sci. Rep. 5 10249 (2015)
Co/Pt	3.4 nm	FMR spin pumping	PRL 112 106602 (2014)
NiFe/Pt	3.7 nm	FMR spin pumping	PRB 83 144402 (2011)
NiFe/Pt	4.0 nm	FMR spin pumping	PRB 88 064414 (2013)

Table R2. Spin diffusion length of W

W (Material structure)	Spin diffusion length	Methods	Reference
CoFeB/W	1.3 nm	Spin Hall magnetoresistance	PRL 116 097201 (2016)
Co/W	1.6 nm	Spin Hall magnetoresistance	APL 107 192405 (2015)
YIG/W	2.1 nm	FMR spin pumping	PRL 112 197201 (2014)
CoFeB/W	2.1 nm	Spin Hall magnetoresistance	Sci. Rep. 5 14668 (2015)
YIG/W	2.1 nm	FMR spin pumping	JAP 117 172603 (2015)
CoFeB/W	3.0 nm	FMR spin pumping	Nature Comm. 7 10644 (2016)

Figure R1 | Effect of the thermal conductivity (κ) of HM materials on the temperature gradient. a,b, Lateral temperature distribution under the edge illumination for W (3 nm)/CoFeB(2 nm)/MgO(1 nm)/Ta(1 nm) sample (a) and Pt(3 nm)/CoFeB(2 nm) /MgO(1 nm)/Ta(1 nm) sample (b). The black lines represent the temperature profiles calculated using the thermal conductivities of $\kappa_{Pt}=72$ W/mK, $\kappa_W=163$ W/mK, and $\kappa_{CoFeB}=10$ W/mK, whereas the red lines are the results obtained by using the same thermal conductivities of the HMs with that of CoFeB, $\kappa_{Pt}=\kappa_W=\kappa_{CoFeB}=10$ W/mK.

Figure R2 | Transverse spin Nernst magnetoresistance in W(3 nm)/CoFeB(2 nm) sample under different magnetic fields of 30 mT, 60 mT, and 100 mT. a-c, Thermoelectric Hall signals for laser illumination at left edge (a), centre (b), right edge (c) of the sample.

Figure R3 | Thickness dependence of transverse SNMR in W/CoFeB and Pt/CoFeB structures. **a, b,** Transverse SNMR $V_{2\theta}$ versus HM thickness for edge illumination in W/CoFeB structure (a) and Pt/CoFeB structure (b). The white circles represent experimental data and solid lines represent best fitted curves, while purple band indicates error ranges of extracted values, which originated from uncertainties of S_{HM} , G , and θ_{SH} .

Figure R4 | Offset in thermoelectric Hall signals of a W(3 nm)/CoFeB(2 nm) sample. The black symbols are raw thermoelectric Hall signals that have a d.c. offset of $\sim 6 \mu\text{V}$. The red symbols represent the angular dependence of the transverse SNMR signals obtained by subtracting the offset.

Figure R5 | Measurement of sample temperature induced by laser illumination. **a**, A schematic illustration of the sample structure with a wire of 1 μm width and 3 μm length. **b**, Temperature dependence of the resistance in Pt(4 nm)/CoFeB(2 nm) sample. **c**, Changes in resistivity and resultant temperature by the laser illumination at different laser position from the centre of the sample.

Figure R6 | Effect of the capping layer of MgO(1 nm)/Ta(1 nm) on temperature profiles. **a-c**, The black (red) lines represent calculated temperature profiles without (with) the capping layer for single CoFeB(2 nm) layer (a), Pt(3 nm)/CoFeB(2 nm) (b), and W(3 nm)/CoFeB(2 nm) (c) samples.

Figure R7 | Thermoelectric signals depending on laser position along the y-axis. **a**, A schematic illustration of the sample structure for the measurement. Each color represents a different laser location along the y-axis. **b**, Thermoelectric Hall signals under centre illumination with different laser locations. **c,d**, Thermoelectric Hall signals under the left edge (c) and the right edge (d) illumination of the laser located at $y = 0$ mm (black) and $y \sim +0.3$ mm (red).

Figure R8 | Normalized transverse spin Nernst magnetoresistance with resistance originating from SNE in various layer structures. **a**, Schematics of measurement under different laser position on bar-shaped structure. The x - z plane view indicates the laser positions along the x direction. Each color of circle represents the laser position. **b-e**, Thermoelectric Hall signals for W(3 nm)/CoFeB(2 nm) (**b**), CoFeB(2 nm) (**c**), Pt(3 nm)/CoFeB(2 nm) (**d**), and Cu(3 nm)/CoFeB(2 nm) structures (**e**) for different laser locations, at the centre ($x \sim 0 \mu\text{m}$, top panel), edge ($x \sim 5 \mu\text{m}$, middle panel), and outside of the structure ($x \sim 10 \mu\text{m}$, bottom panel) for each sample. Dotted and dash-dotted lines in the middle panel show the decomposition of two angle-dependent signals of $\cos\theta$ and $\sin 2\theta$. The symbol colors denote the laser position as illustrated in schematics of Fig. 2a.

Figure R9 | Angular dependence of the electrical magnetoresistance in W(2 nm)/CoFeB(2 nm) sample. Angular dependence of the longitudinal resistance R_{xx} (a, b, c) and transverse resistance R_{xy} (d, e, f) of the W(2 nm)/CoFeB(2 nm) sample for α , β , and γ scan, respectively. The measurements were done under a magnetic field of 9 T.

Figure R10 | Angular dependence of the electrical magnetoresistance in W(4 nm)/CoFeB(2 nm) sample. Angular dependence of the longitudinal resistance R_{xx} (a, b, c) and transverse resistance R_{xy} (d, e, f) of the W(4 nm)/CoFeB(2 nm) sample for α , β , and γ scan, respectively. Same angular experiments were done as shown in Fig. R9.

Figure R11 | Angular dependence of the electrical magnetoresistance in W(5 nm)/CoFeB(2 nm) sample. Angular dependence of the longitudinal resistance R_{xx} (a, b, c) and transverse resistance R_{xy} (d, e, f) of the W(2)/CoFeB(2) sample for α , β , and γ scan, respectively. Same angular experiments were done as shown in Fig. R9.

Figure R12 | Thickness dependence of transverse SNMR and spin Hall magnetoresistance in W/CoFeB structure

Reviewers' Comments:

Reviewer #1 (Remarks to the Author):

The authors have satisfactorily addressed all of the concerns that I raised on the first submission. Therefore, I recommend this revised manuscript for publication in Nature Communications as is.

Reviewer #2 (Remarks to the Author):

I am pleased to find that the authors did a great job to clarify my questions. I can recommend to publish the paper in the present form.

Reviewer #3 (Remarks to the Author):

In this revised manuscript the authors Park et al. have largely addressed the comments and suggestions. The authors have made necessary and appropriate revisions. After carefully reviewing all of the replies together with the manuscript, I recommend the publication of this manuscript in Nature Communications as it is.

One comment suggesting the concerns from reviewer #2 about the novelty of this work: There are several works on arXiv about the observation of the SNE at the moment. However, many of them cannot properly demonstrate the real existence of the SNE. Furthermore, the experimental methods differ, in the temperature gradient generation, for instance, and the separation or even concerning of unintended effects. Nature Communications' aim is to publish „important advances of significance to specialists“. In my opinion, this is factual for this work. It is important to promote significant work in high rank journals.